# Analysis of High-Order Surface Gratings Based on Micron Lasers on Silicon

Jiachen Tian [1,2], Licheng Chen [1,2], Xuliang Zhou [1,*], Hongyan Yu [1], Yejin Zhang [1,2] and Jiaoqing Pan [1,2,*]

1    Key Laboratory of Optoelectronic Materials and Devices, Institute of Semiconductors, Chinese Academy of Sciences, Beijing 100083, China; jctian@semi.ac.cn (J.T.); lichengchen@semi.ac.cn (L.C.); hyyu09@semi.ac.cn (H.Y.); yjzhang@semi.ac.cn (Y.Z.)

2    Center of Materials Science and Optoelectronics Engineering, University of Chinese Academy of Sciences, Beijing 100049, China

*    Correspondence: zhouxl@semi.ac.cn (X.Z.); jqpan@semi.ac.cn (J.P.)

**Abstract:** High-quality silicon-based lasers are necessary to achieve full integration of photonic and electronic circuits. Monolithic integration of III–Vmicron lasers on silicon by means of the aspect ratio trapping (ART) method is a promising solution. To obtain sufficient optical feedback to excite the laser without introducing complex fabricating processes, we have designed a high-order surface grating on micron lasers which was epitaxially grown by the ART method and can be fabricated by common UV lithography. The performance of the grating was analyzed by the finite-difference time-domain (FDTD) method and eigenmode expansion (EME) solver. After simulation optimization, the etching depth was set to 0.6 μm to obtain proper reflection. The width of the slots and the slot spacing were selected to be 1.12 μm and 5.59 μm, respectively. Finally, we obtained results of 4% reflectance and 82% transmittance at a 1.55 μm wavelength at 24 periods.

**Keywords:** high-order grating; surface grating; aspect ratio trapping; finite-difference time-domain (FDTD); eigenmode expansion (EME)



## 1. Introduction

With the increasing demand for information processing capabilities and transmission speed in modern society, opto-electronic integrated circuit (OEIC) technology has received significant attention. Photons have superb features in terms of transmission speed, bandwidth, and energy consumption, which makes them a promising solution to replace electrons in modern communications [1]. Silicon-based passive devices, including waveguide devices, detectors, modulators, resonant cavities, etc., have made significant advances. However, a silicon-based electrically pumped laser is still an unsolved problem because silicon is an indirect bandgap material with low luminescence efficiency [2]. Since III–V materials have highly efficient optical properties, the heterogeneous epitaxy of III–V materials on silicon provides a reliable solution to the silicon-based laser problem.

There are many significant issues when directly epitaxially growing III–V materials on silicon substrates. Since there are non-negligible differences in lattice constants, thermal expansion coefficients, and polarity between silicon and III–V materials, a large number of dislocations are generated when heterogeneously epitaxial III–V materials are grown on silicon substrates. To fabricate reliable lasers on silicon [3], quantum dot (QD) [4–6], bonding [7,8], and aspect ratio trapping (ART) methods [9] were introduced. The benefits and drawbacks of these methods are shown in Table 1. The ART technique is to grow heterogeneous materials epitaxially in a trench with a depth-to-width ratio greater than 1 in order to confine defects to the trench. However, electrically pumped lasers at 1.55 μm made with the ART method have not yet been realized due to the significant lattice mismatch between InP and Si, making it difficult to grow high-quality InP structures epitaxially [3]. The production of high-quality lasers using the ART method remains appealing for industrial

applications, as it enables large-scale silicon-based heterogeneous integration. Therefore, further research in this area is needed.

**Table 1.** The benefits and drawbacks of quantum dots, bonding, and ART.

| Methods | Bonding | Quantum Dots | Aspect Ratio Trapping |
|---|---|---|---|
| Commercialization | Already Commercialized | Under Research | Under Research |
| Coupling | Needs to be Aligned | Hard | Easy |
| Electronically Pumped At C-band | Already Realized | Already Realized | Not Realized |
| Buffer Layer | No Buffer Layer | Thick | Thin |

In the previous work, to solve the misfits, threading dislocations, and antiphase domain boundaries (APBs) caused by the differences between Si and III–Vmaterials, the ART method, two-step growth process, and V-grooves were introduced into the epitaxy process [10].

The ART technique has evolved from the "epitaxial necking method" proposed by T.A. Langdo et al. in 2000 [11]. Initiating material was first grown inside Si trenches confined by oxide spacers. Crystal epitaxy would energetically prefer to occur at the bottom of the Si trenches rather than on the surface of SiO$_2$. The inclined threading dislocations generated from the bottom Si interface would terminate at the oxide sidewalls when the depth-to-width ratio of the trench is greater than 1.

In recent research, we obtained the structure of an InP microwire with a reversed ridge waveguide (RRW) epitaxially grown on a silicon substrate by metal–organic chemical vapor deposition (MOCVD) using the aspect ratio trapping (ART) method [12]. InGaAs/InGaAsP multi-quantum-well (MQW) structures with InGaAsP separate confinement heterostructures (SCH) were introduced into InP microwires. For better implementation of silicon-based electrically pumped lasers, distributed feedback (DFB) lasers have attracted our attention.

Traditional distributed feedback (DFB) lasers use Bragg gratings for optical feedback to achieve single mode with high quality, allowing better feedback amplification at specific wavelengths. These gratings are usually etched close to the active layer and need to be buried by secondary epitaxy [13]. However, this increases the complexity of the process and reduces the yield and reliability of the device. To reduce the cost and difficulty of device fabrication, our structure uses surface grating, eliminating the complex steps required for secondary epitaxy [14–17]. In addition, the size of the first-order Bragg grating is very small, usually a few hundred nanometers, which is difficult to fabricate. To reduce the fabrication difficulty and cost, we use a high-order grating with a grating slot width larger than 1 micron, which can be achieved by general lithography.

In this paper, the finite-difference time-domain (FDTD) method and eigenmode expansion (EME) solver are used to analyze the high-order slotted surface gratings designed on the structure of an InP microwire with reversed ridge waveguide (RRW) on a silicon substrate. Simulations show that the power reflection and transmission of the micron laser change abruptly at a slot depth of 0.6 μm. High-order surface gratings at 1.55 μm with low loss were achieved in the simulation. The slot width is set to 1.12 μm, and the slot spacing is set to 5.59 μm. At a period of 24, the reflectance at 1.55 μm is about 4%, and the transmittance is about 82%.

## 2. Laser Structure and Simulation Model

The structure of an InP microwire with reversed ridge waveguide (RRW) on silicon substrate was obtained by using MOCVD (AIXTRON 200, Herzogenrath, Germany). The model we set for simulation is shown in Figure 1a. The structure was grown on the pre-patterned Si. The origin of the vertical scale is on the top of the silicon substrate. The width of the SiO$_2$ trench is 500 nm and the height of the trench is 1 μm. GaAs was grown in a V-groove which is formed by two {111} facets of Si. The InP out of the SiO$_2$ trench

is formed by two {111} facets and three {001} facets. The top {001} facet is nanoscale. In the simulation, we set the height and width of the InP structure out of the SiO$_2$ trench to 1.8 μm and 2 μm, respectively, according to the actual growth structure. The thickness of the InGaAs/InGaAsP multi-quantum-well (MQW) structure with InGaAsP separate confinement heterostructure (SCH) was 300 nm in total, and the refractive index was set at 3.575 [18]. After simulation by the FDTD method, the result is the optical transverse mode (TE$_{00}$) supported by the microwire, as shown in Figure 1b. Other modes calculated by FDTD Solutions have significant leakage into the silicon substrate. The TE$_{00}$ mode distribution demonstrates that the RRW structure is able to confine the light field to the active region with MQW outside the silica trench, effectively avoiding the leakage of the light field to the silicon substrate.

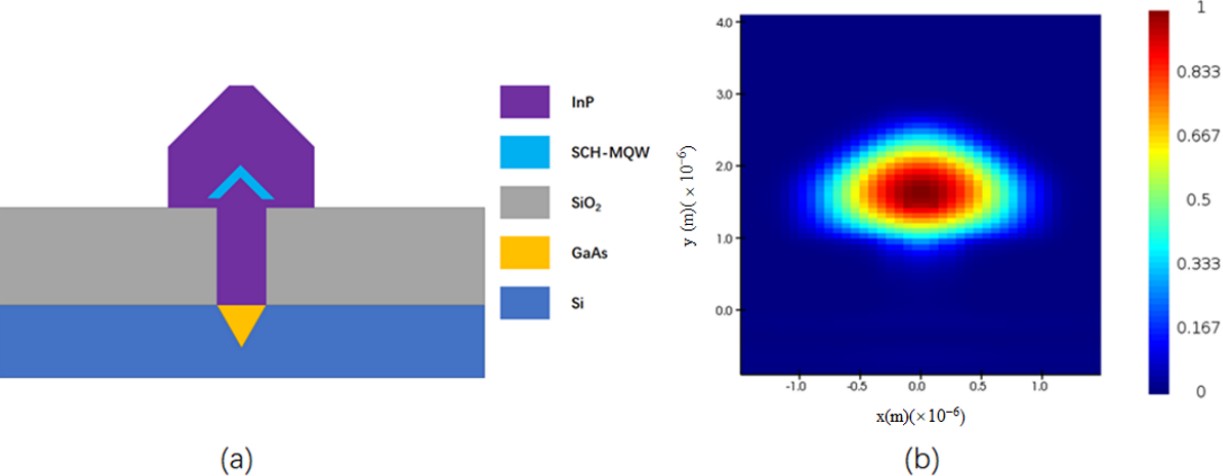

**Figure 1.** (**a**) The crosssectional schematic of the InP microwire with reversed ridge waveguide (RRW) on silicon substrate. (**b**) Optical mode distribution in microwire.

The cross-sectional view of a silicon-based high-order surface grating laser along the transmission direction is shown in Figure 2. In the figure, the $d_s$ and $d_w$ are the slot width and slot spacing between two slots. The width of the slot $d_s$ can be defined as

$$d_s = \frac{(2p+1)\lambda}{4n_s},\tag{1}$$

and the width of the slot spacing $d_w$ can be defined as

$$d_w = \frac{(2q+1)\lambda}{4n_w}\tag{2}$$

where $\lambda$ is the wavelength in air, $n_s$ and $n_w$ are the effective index of the slot and slot spacing, and $p$ and $q$ are integers. The period of the Bragg grating $\Lambda$ is given by the following equation:

$$\lambda = d_s + d_w,\tag{3}$$

and the order of the Bragg grating $m$ can be expressed as

$$m = p + q + 1.\tag{4}$$

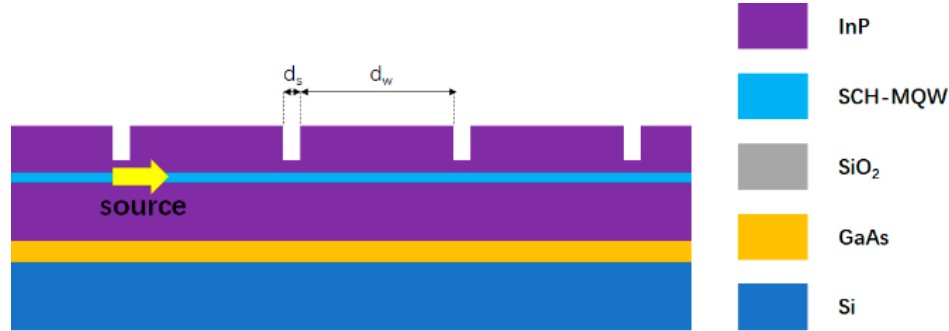

**Figure 2.** The crosssectional view of a silicon-based high-order surface grating laser along the transmission direction.

### 3. Simulations

*3.1. Single Slot Structure Simulated with 3D FDTD*

The RRW structure with a single slot was analyzed first to investigate the effect of single slot depth on amplitude reflection and transmission. To show the parameters of the simulation more clearly, the details about the simulations by FDTD and EME are shown in Table 2. A uniform mesh with step size of 0.00025 μm was used in the simulation. The simulation time was 3000 fs and the simulation temperature was at 300 K. PML absorbing boundaries were applied to four sides of the simulation window. The source was selected to Mode Source and the wavelength of the source was set to 1550 nm. The width of the slot was set to 1.12 μm. The background index was set the same as SiO$_2$. We added a power monitor behind the slot to observe transmission and a monitor in front of the slot to observe reflection. In the simulation, the transmittance monitor was placed at a position 80 μm behind the trench, the reflectance monitor was placed at a position 80 μm in front of the trench, and the Mode Source was set at 75 μm in front of the trench, scanning the depth of the slot from 0 to 1 μm. The length of the waveguide in the simulation region was 180 μm. The size of Mode Source covered the whole laser cross-section. The mode profile was selected near the SCH-MQW structure. Results are shown in Figure 3a. The effect on transmission T and reflection R is very small until the depth of the slot is 0.5 μm, because the mode distribution of the RRW structure is limited in the SCH-MQW part of the structure, which has less effect on mode transmission at smaller depths. When the trench is deeper than 0.5 μm, the transmission starts to drop and the reflection starts to rise significantly, at which time the influence of the slot on the modes starts to become larger and the reflection on the power transmission is apparently enhanced. As shown in Figure 1b, the very top of the structure is at 2.8 μm in the y direction. When the trench depth is beyond 0.5 μm, the bottom of the trench begins to coincide with the main distribution of the mode pattern. At this point, the mode distribution moves downward and a more pronounced light leakage into the silicon substrate can be detected, as shown in Figure 3b. Therefore, the transmission starts to drop sharply.

**Table 2.** The details about the simulations by FDTD and EME.

| Parameters | Simulations by FDTD | Simulations by EME |
|---|---|---|
| Time | 3000 fs | 3000 fs |
| Source Type | Mode Source | Mode |
| Mesh Size | 5 μm × 5 μm × 180 μm | 4 μm × 3 μm × ($d_s$ + $d_w$) × number of periods |
| Mesh Step | $2.5 \times 10^{-4}$ μm | $1 \times 10^{-6}$ μm |
| Simulation Temperature | 300 K | 300 K |
| Boundary Conditions | PML | PML |

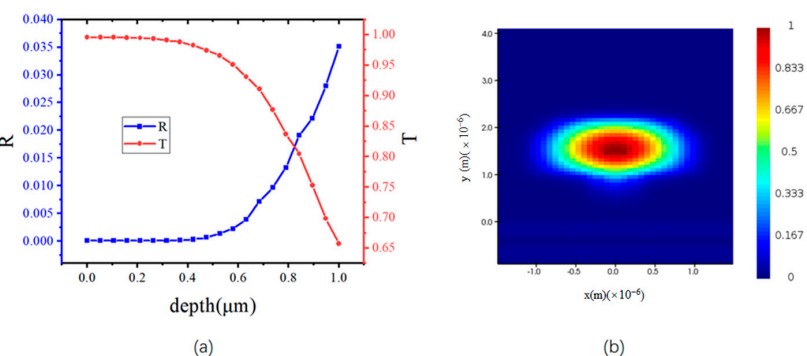

**Figure 3.** (**a**) The amplitude transmission and reflection of a single slot versus the slot depth analyzed by 3D FDTD. (**b**) Optical mode distribution in microwire when the depth of the slot is 0.5 μm.

Considering that the surface grating is a multi-slot structure, we needed the slot structure to provide a suitable amplitude reflectivity without introducing excessive losses, so we chose a slot depth of 0.6 μm for the next step of simulation optimization.

### 3.2. Single Slot Structure Simulated with the EME Method

Furthermore, we used the EME solver to simulate the transmission, reflection, and loss of the high-order surface gratings. The EME method discretizes the structure along the propagation direction using cells and solves for the modes supported by each cell. This approach allows the EME method to scale exceptionally well with respect to propagation distance, making it an ideal choice for simulating long structures. In contrast, FDTD-based methods may not be as well-suited for this purpose.

First, a single slot was analyzed. The periodic group was set, including a slot and a waveguide as slot spacing. To analyze the influence of the width of the slots, the number of periodic groups m was set to 1. The depth of the slot was kept at 0.6 μm and the wavelength was 1.55 μm. The variation of reflection and transmission versus the slot width is shown in Figure 4. It can be seen that the transmittance of light from a single slot has an overall decreasing trend as the slot width increases. The periodicity of fluctuations roughly satisfies $\lambda/2n_s$, which can be explained by (1). The oscillation of the power reflection is caused by the FP resonance between the two interfaces of the slot. The first interface scatters the guided mode into the slot region. The scattered field keeps propagating forward and then is reflected back by the second interface. Since we expect lower loss in the grating, the narrower slots are preferred. When the mode gain is greater than the sum of the losses in the laser, the laser reaches the threshold condition and achieves excitation. In order to reduce the cost and difficulty of device preparation, we choose a grating feature size larger than 1 μm, which can be realized by common UV lithography techniques. According to (1), when *p* is taken to be 4, the width of the slot is set to 1.12 μm.

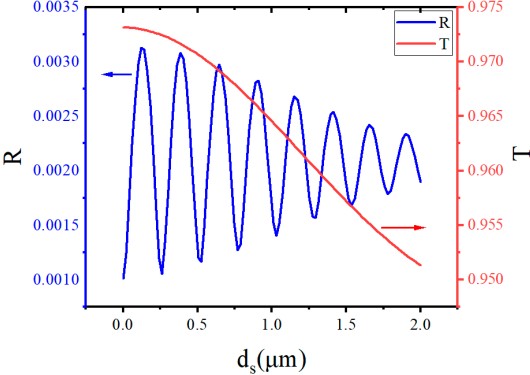

**Figure 4.** The variation of reflection and transmission versus the slot width of the single-slot structure simulated with the EME method. The arrow points to the vertical axis that corresponds to the curve.

### 3.3. Multi-Slot Structure Simulated with the EME Method

Then we focused on the multi-slot structure. In order to achieve a higher reflectivity of a Bragg grating laser, we needed a high number of grating cycles, which requires a relatively long cavity length. However, long cavity lengths are prone to defects and reduced yield. This laser is able to get excitation after CrossLight simulation with a waveguide length around 200 μm. Considering this, the cavity length was designed to be around 200 μm.

The slot depth was fixed at 0.6 μm, the width of the slots was set at 1.12 μm, and the transmission wavelength was kept at 1.55 μm. The distance between the slots was varied to observe the effect of the distance between the slots on transmittance and reflectance. High reflectivity requires more periods, which can lead to longer cavity length and a decrease in yield rate. To control the length of the grating to get enough reflectivity with high yield, the number of periods was set to 24. Figure 5 indicates that the transmittance of the grating peaks at spacings of around 5.5 μm and 7 μm, where the transmissions are both over 80%. In order to get more grating periods in a certain cavity length, and thus obtain higher reflectivity, shorter spacing was chosen. Here, we preferred the width of the spacing near 5.5 μm. Combined with (2), we finally selected q as 22, and the length of the slot spacing as 5.59 μm.

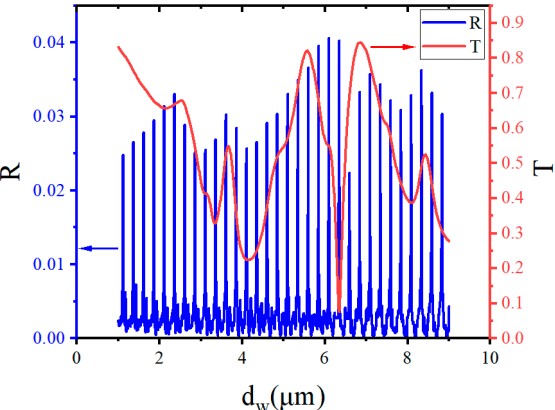

**Figure 5.** The variation in reflection and transmission versus the width of the slot spacing between two slots of the multi-slot structure with the EME method. The arrow points to the vertical axis that corresponds to the curve.

When the width of the slot spacing was fixed at 5.59 μm, the effects of the width of the slots on transmission and reflection were also explored. The results are shown in Figure 6. The reflection reaches the peak periodically, and the interval of the peak is satisfied at $\lambda/2n_s$. The transmission exceeds 80% at slot widths of 1 μm and 3 μm.

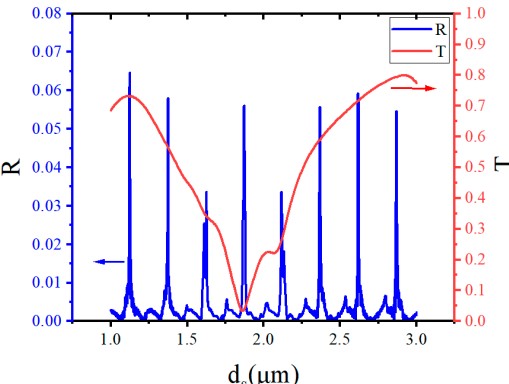

**Figure 6.** The variation in reflection and transmission versus the width of the slots of the multi-slot structure with the EME method. The arrow points to the vertical axis that corresponds to the curve.

In order to obtain more comprehensive effects of the slot width and the width of the slot spacing on the transmission of the grating, we made a contour plot of the influence of $d_s$ and $d_w$ on transmission. According to Figure 7, there are two areas with high transmittance, and both can obtain a transmission of over 80%. The parameters needed in the grating should satisfy a grating feature size larger than 1 μm, short cavity length, low loss, and high reflection. Taking it all into consideration, the gratings with an etching depth of 0.6 μm, slot width of 1.12 μm, and slot spacing of 5.59 μm were selected. By (4), the order of the Bragg grating is 27.

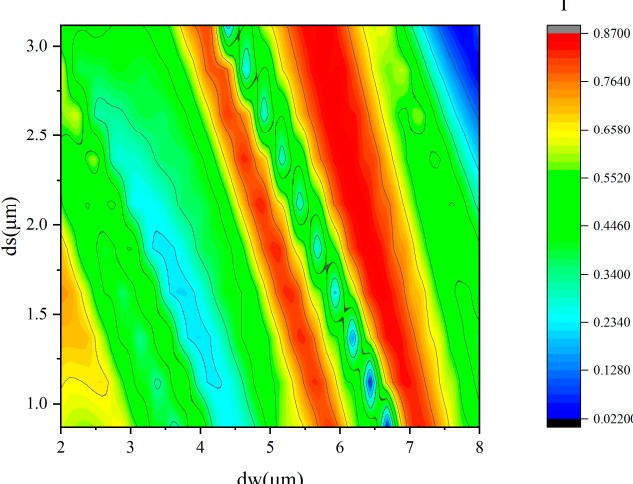

**Figure 7.** The contour plot of the influence of the width of the slots and the distance between slots on transmission.

After the parameters of the high-order surface gratings were determined, the wavelength of the source was scanned to obtain the reflection spectrum. The peak of the reflection wavelength is perfectly consistent with the Bragg equation. In the design of the grating structure, at the period number of 24, the reflectivity of the 27th surface grating at a wavelength of 1.55 μm was about 4% and the transmittance was around 82%, as shown in Figure 8. The maximum reflectivity obtained here is different from that obtained for similar parameters in Figure 6. This is due to the fact that the value of $d_s$ in Figure 8 is taken to be 1.12 μm, and not the more accurate value that achieves the maximum reflectivity in Figure 6. This indicates that R is very sensitive to $d_s$.

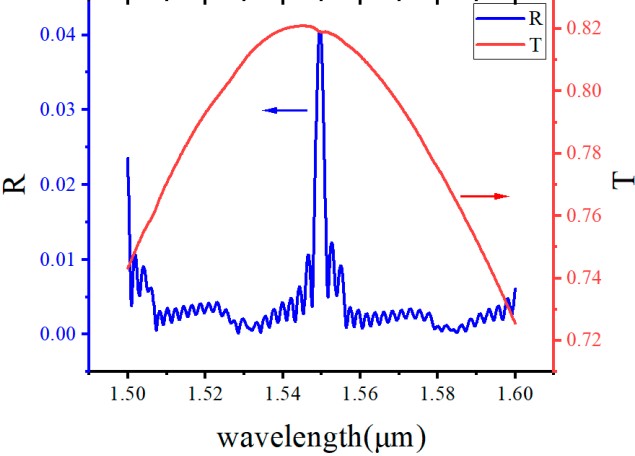

**Figure 8.** The variation in reflection and transmission versus the wavelength of the source of the multi-slot structure with the EME method. The arrow points to the vertical axis that corresponds to the curve.

During the fabrication of the laser, it may be required to achieve a higher reflectivity; hence, the number of cycles should be controlled and the trade-off between high reflectivity and losses should be made. As shown in Figure 9, the reflectivity of the grating is more than 10% but the corresponding transmittance is less than 60% when the number of periods is 100, at which time the length of the laser is close to 700 µm. Since the reflection coefficient of the grating of an efficient high-power DFB laser should be less than $10^{-1}$ [19,20], The period of the grating must not exceed 100.

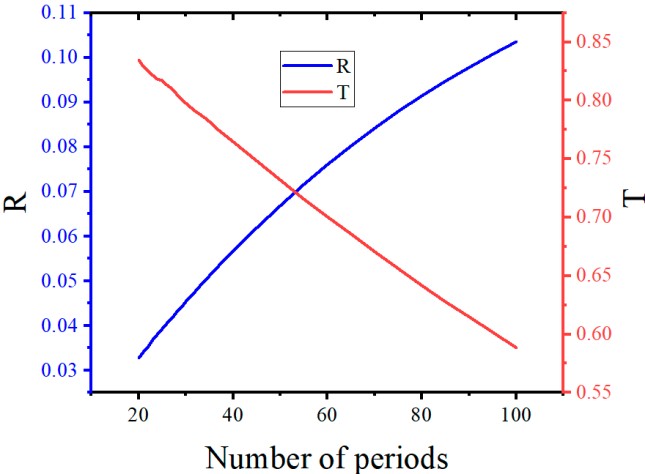

**Figure 9.** The curves of refractive index and transmittance at period numbers from 20 to 100 with $d_s$ = 1.12 µm and $d_w$ = 5.59 µm.

### 3.4. Final Verification of High-Order Surface Gratings by FDTD

Parameters obtained by EME simulation are then introduced into FDTD Solutions to observe the propagation of the light at 1.55 µm. The optical field distribution along a 160-µm propagation length inside the sub-micron laser viewed in the XZ plane is shown in Figure 10a. The light at 1.55 µm can be effectively reflected by the high-order surface gratings and the can propagate with low loss. By the optical field distribution along a 160-µm propagation, we found that after 160 µm of propagation, the optical field distribution changes because of the scattering of light. The $TE_{00}$ mode is also affected by the gratings. Along the transmission direction, the $TE_{00}$ mode at 160 µm is shown in Figure 10b, which is different from the $TE_{00}$ mode at the beginning. The transmittance and refractive index calculated with FDTD are in close agreement with the values calculated with EME.

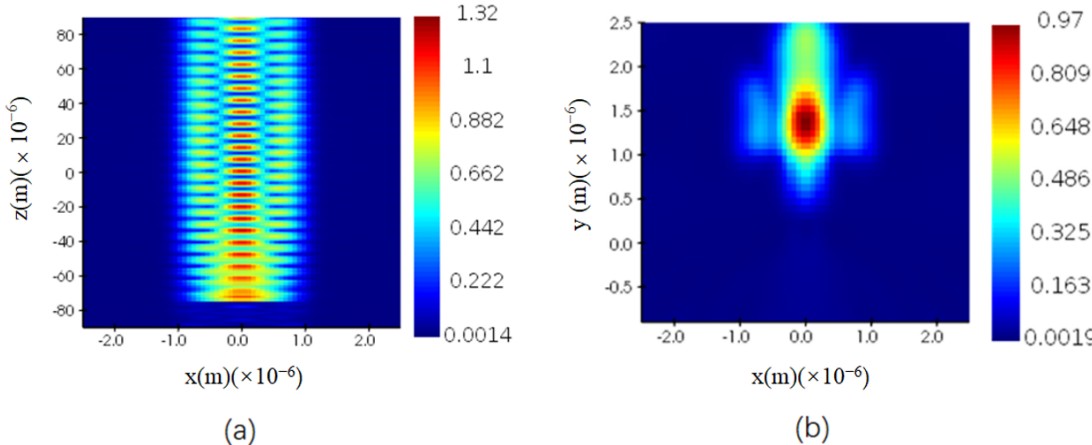

**Figure 10.** (**a**) The optical field distribution along a 160-µm propagation length inside the sub-micron laser, viewed in the XZ plane (**b**) The optical mode distribution after 160-µm propagation, viewed in the XY plane.

*3.5. Fabrication Tolerance Analysis*

As discussed above, relatively good transmittance with low loss can be obtained at $d_s$ = 1.12 μm and $d_w$ = 5.59 μm. In Figure 7, we are able to obtain lower losses over the range of the wider red region. However, the depth of etching slots during fabrication can also have a large influence on the performance of the device. The curves of refractive index and transmittance at slot depths from 0.5 μm to 0.7 μm at 24 periods were calculated using the EME method with $d_s$ = 1.12 μm and $d_w$ = 5.59 μm. The results are shown in Figure 11. Here, we take the values for the highest reflectance and transmittance near 1.55 μm. As expected, the reflection increases, and the transmittance decreases, as the etching depth increases. Here, we want to control the reflectivity over 0.03 while the transmission is over 0.75 to ensure the high efficiency of the grating. According to the figure, the tolerance needs to be limited to ±30 nm.

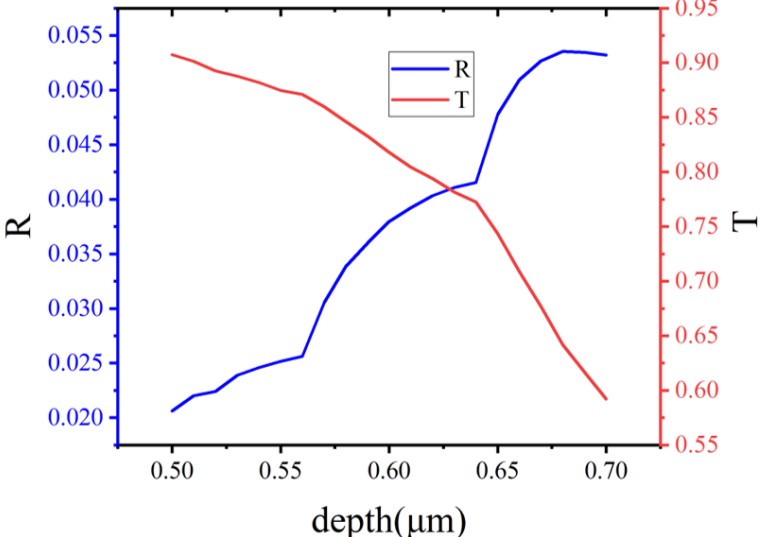

**Figure 11.** The amplitude transmission and reflection of the multi-slot structure versus the slot depth analyzed by the EME method.

## 4. Conclusions

Based on the RRW structure, a high-order surface grating was designed for better optical feedback and easy fabrication. Though the simulation result of a single-slot structure with the 3D FDTD method, the depth of the slots was set to 0.6 μm. Then, to save computational resources, the EME method was used for further simulation. The transmission wavelength was fixed at 1.55 μm; considering the loss and the difficulty of fabrication, the grating slot width of 1.12 μm and slot spacing of 5.59 μm were selected. With the number of periods at 24, the reflectivity of 1.55 μm wavelength was about 4% and the transmittance was around 82%. This approach might provide a method for silicon-based single-mode or even multi-wavelength lasers.

**Author Contributions:** Conceptualization, J.T. and X.Z.; methodology, J.T. and X.Z.; software, J.T. and L.C.; validation, J.T., H.Y., Y.Z. and X.Z.; formal analysis, J.T.; investigation, J.T. and X.Z.; resources, X.Z. and J.P.; data curation, J.T.; writing—original draft preparation, J.T.; writing—review and editing, J.T., X.Z. and J.P.; visualization, J.T.; supervision, J.T. and J.P.; project administration, X.Z. and J.P.; funding acquisition, X.Z. and J.P. All authors have read and agreed to the published version of the manuscript.

**Funding:** This study was funded by the Strategic Priority Research Program of the Chinese Academy of Sciences (Grant No. XDB43020202), and the Beijing Natural Science Foundation (Grant No. Z200006).

**Institutional Review Board Statement:** Not applicable.

**Informed Consent Statement:** Not applicable.

**Data Availability Statement:** Data are contained within the article.

**Conflicts of Interest:** The authors declare no conflicts of interest.

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
