# Peer review of "Analysis of High-Order Surface Gratings Based on Micron Lasers on Silicon"

_photonics, doi:10.3390/photonics11010092_

Round 1

Reviewer 1 Report

Comments and Suggestions for Authors

The work titled "Analysis of High-order Surface Grating Based on Submicron Lasers on Silicon" shows simulation results on high-order surface gratings on submicron lasers developed by aspect ratio trapping (ART) method. This work is of great value for the realization of silicon-based electrically pumped lasers. The results are presented and the manuscript is well organized.

Below the authors can find comments and questions that need to be addressed to improve the quality of the manuscript:

(1)   Different fabrication techniques (quantum dots, bonding, ART) are only cited in the introduction. The difference between these techniques is not clear. A table could be added in this section to compare the benefits and drawbacks of these methodologies.

(2)   The details about the simulations are not clear. The Authors should report the main parameters used for the simulations (i.e.: time, mesh size, source type, ...). Authors should also consider adding a table to summarize the main parameters used for the different simulations.

(3)   More details need to be shown to explain why using EME method for simulations in Section 3.

(4)   In Section 3.4, some formats have not been corrected. For example: ‘TE00’ should be changed to ‘TE00’.

 Overall, the authors have presented a high-order surface grating with promising performance metrics which can be fabricated easily. This work can contribute to the realization of silicon-based lasers in the C-band. I recommend this manuscript for publication in Photonics after some language improvement and explanation.

Comments on the Quality of English Language

Minor editing of English language required

Author Response

Thank you very much for your guidance on this article.

We appreciate that your suggestions are very constructive, and we have responded to each of them based on your comments and have made additions and changes to the original text, which can be found in the attachments.

Reviewer 2 Report

Comments and Suggestions for Authors

This manuscript prepared by Jiachen et al. aimed to design high-order surface grating structures to improve silicon-based on-chip laser. The theoretical analysis is designed finely, but the introduction needs to be more carefully written, at least avoiding typos and grammar mistakes. My comments are provided below. 

1. The author should explicitly define and describe ART technique in detail at an early position of introduction to improve the readability.  Starting from line 49, reference is also needed. 

2. Line 40 to 43 are hard to read.

3. With all the sizes and geometries larger than 1um, why the authors consider their design 'submicron'?

4. The author may want to add more information, such as the electric field profile, to explain their finding in Fig3, where the transmission starts to drop when the trench depth is beyond 500nm. 

5. What is the size of the mode source and what is the exact position in Fig3 simulation? If this proposed system is converted into real application, how would it be pumped? If optically pumped, will the light be sent through another waveguide and how would it couple into the structure in this manuscript? If it would be electrically pumped, will it be more accurate to use a dipole source as an excitation?

6. The authors may want to explain how this 4% reflection can be used as a feed back cavity for the laser.

Comments on the Quality of English Language

Many typos and grammar mistakes need to be corrected.

Author Response

Thank you very much for your guidance on this paper.

We considered your suggestions to be very constructive, and we have responded to each of them based on your comments, and have made additions and changes to the original text, please see the attachments.

Reviewer 3 Report

Comments and Suggestions for Authors

My comments and suggestions as well as a figure are included in the attached pdf.

Author Response

(The authors gave the same response as above.)

Round 2

Reviewer 2 Report

Comments and Suggestions for Authors

The authors have addressed my comments and concerns. The language is also improved. 

Author Response

Thank you for your suggestions and affirmations, this article has become more complete and convincing because of your suggestions. Our subsequent experiments are being organized. More simulations will be based on the results of later experiments.

Reviewer 3 Report

Comments and Suggestions for Authors

The paper is all in all ok.

The English is much improved.

A good portion of the suggestions has also been implemented, including the sensitivity to manufacturing tolerances with respect to groove depth (new Figure 11). However, there seems to be a small error in the explanation (line 271): As I understand it, in Fig. 11, the depth was varied for a fixed number of grooves. If this is the case, please specify the number of grooves used in the simulation in the figure caption.

Regarding tolerances, the "coverletter" mentions (in a different context) that the sensitivity to slot width ds is actually quite strong: A change in ds of only 2.12 nm (0.2 % of ds) changes the reflectance by 50 % from R = 4 % to R = 6.118 %.

I also appreciate the additional reference by Wenzel et al. The parameters there (and in ref. 7 by Decker et al. therein) refer to DBR lasers with lengths of approximately 5 mm.  However, I am not sure whether these parameters can be applied 1 to 1 to the lasers discussed here with lengths of approximately 200 µm.

By the way, both, the Wenzel and the Decker papers include the simulation of the passive grating waveguide, the simulation of the laser device, and experimental laser results. But that was almost a decade ago and does not seem to be the way papers are published nowadays. Let’s see whether there will be two more follow-up papers clarifying the laser properties by simulation and experiment, respectively.

Author Response

Thank you very much for your affirmation and suggestions, subsequent experiments have been steadily scheduled and initiated.

(1)I'm very sorry for the confusion caused by my oversight, the description of the period change was for Figure 9 and I mistakenly pasted the description into the explanation of Figure 11. The description has changed to 'The curves of refractive index and transmittance at slot depth from 0.5 μm to 0.7 μm at the number of periods of 24'.  

(2)Thanks to your suggestion, we have put the description from the 'coverletter' into the paper in line 238 to enhance the reader's understanding of the structure.

(3)Our work is just a simulation in advance and will be adjusted after the experiment based on the data. More simulations will be based on the results of later experiments. After the epitaxial quality is improved, we will consider longer laser cavities. 

(4)Thank you again for your confidence. We will include simulation and experimentation of lasers in our subsequent work and update our simulation and experimentation methods.